# Multiscale Quantization for Fast Similarity Search

**Xiang Wu  Ruiqi Guo  Ananda Theertha Suresh  Sanjiv Kumar**
**Dan Holtmann-Rice  David Simcha  Felix X. Yu**

Google Research, New York
{wuxiang, guorq, theertha, sanjivk, dhr, dsimcha, felixyu}@google.com

## Abstract

We propose a multiscale quantization approach for fast similarity search on large, high-dimensional datasets. The key insight of the approach is that quantization methods, in particular product quantization, perform poorly when there is large variance in the norms of the data points. This is a common scenario for real-world datasets, especially when doing product quantization of residuals obtained from coarse vector quantization. To address this issue, we propose a multiscale formulation where we learn a separate scalar quantizer of the residual norm scales. All parameters are learned jointly in a stochastic gradient descent framework to minimize the overall quantization error. We provide theoretical motivation for the proposed technique and conduct comprehensive experiments on two large-scale public datasets, demonstrating substantial improvements in recall over existing state-of-the-art methods.

## 1 Introduction

Large-scale similarity search is central to information retrieval and recommendation systems for images, audio, video, and textual information. For high-dimensional data, several hashing based methods have been proposed, including randomized [19, 1, 32] and learning-based techniques [34, 35, 15]. Another set of techniques, based on quantization, have become popular recently due to their strong performance on real-world data. In particular, *product quantization* (PQ) [12, 20] and its variants have regularly claimed top spots on public benchmarks such as GIST1M, SIFT1B [20] and DEEP10M [3].

In product quantization, the original vector space is decomposed into a Cartesian product of lower dimensional subspaces, and vector quantization is performed in each subspace independently. *Vector quantization* (VQ) approximates a vector $x \in \mathbb{R}^{\dim(x)}$ by finding the closest quantizer in a codebook $\mathbf{C}$:

$$\phi_{VQ}(x; \mathbf{C}) = \underset{c \in \{\mathbf{C}_j\}}{\arg\min} \|x - c\|_2$$

where $\mathbf{C} \in \mathbb{R}^{\dim(x) \times m}$ is a vector quantization codebook with $m$ codewords, and the $j$-th column $\mathbf{C}_j$ represents the $j$-th quantizer. Similarly, product quantization (PQ) with $K$ subspaces can be defined as following concatenation:

$$\phi_{PQ}(x; \mathcal{S} = \{\mathbf{S}^{(k)}\}) = [\phi_{VQ}(x^{(1)}; \mathbf{S}^{(1)}); \cdots ; \phi_{VQ}(x^{(K)}; \mathbf{S}^{(K)})] \tag{1}$$

where $x^{(k)}$ denotes the subvector of $x$ in the $k$-th subspace, and $\mathbf{S}^{(k)} \in \mathbb{R}^{\dim(x^{(k)}) \times l}$ is a collection of $K$ product quantization codebooks, each with $l$ sub-quantizers.

Product quantization works well in large part due to the fact that it permits asymmetric distance computation [20], in which only dataset vectors are quantized while the query remains unquantized. This is more precise than techniques based on Hamming distances (which generally require hashing

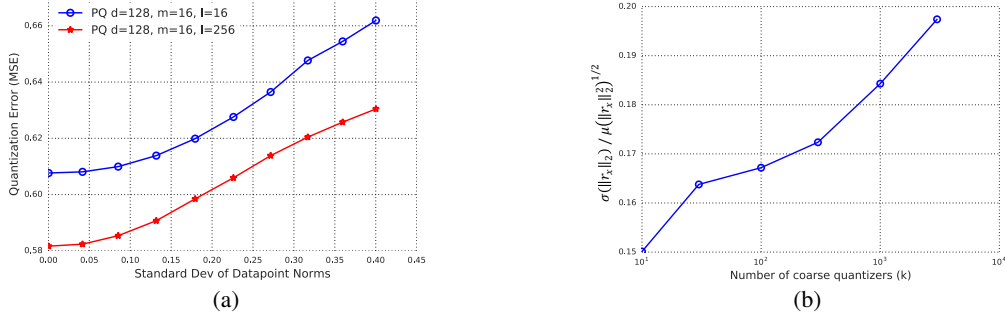

Figure 1: Variance in data point norms poses a challenge to product quantization. (a) PQ quantization error on a *synthetic* dataset $X \in \mathbb{R}^{d \times N}$ grows as the standard deviation of data point norms $\sigma(\|x\|_2)$ increases. The mean of the dataset is zero $\mu(x) = 0$, and the average squared norm is fixed, $\mu(\|x\|_2^2) = 1$. In both settings, $m = 16$ codes are generated per data point and one with $l = 16$ sub-quantizers per subspace, the other with $l = 256$. (b) Ratio between the standard deviation $\sigma(\|r_x\|_2)$ and normalization factor $\sqrt{\mu(\|r_x\|_2^2)}$, where $r_x$ represents the residual after vector (coarse) quantization on the *real-world* dataset of SIFT1M.

the query), while still being efficient to compute using lookup table operations. We will give a more detailed background on product quantization variants in Section 1.2.

## 1.1 Motivation of Multiscale

Empirically, product quantization works the best when the variance in each subspace is roughly balanced [20]. To ensure this, a rotation matrix is often applied to the data prior to performing quantization. This rotation can be either random [20] or learned [11, 30, 39].

In this work, however, we argue that the quality of the product quantization codebook also degenerates when there is variance in the norms of the data points being encoded–even when the variance is relatively moderate. We illustrate this point by generating synthetic datasets such that: (1) the dataset mean is zero; (2) data point direction is chosen uniformly at random; (3) the average squared norm of the data points is fixed. In Figure 1a, we plot quantization error (MSE) of product quantization against the standard deviation of the norms of the data points. Clearly, quantization error increases with the variance of the data point norms. In real-world settings (Figure 1b), the residuals of a coarse vector quantization of the data also tend to have highly varying norms.

To compensate for the case when there is large variance in norms, we modify the formulation of product quantization by separately scalar quantizing data point norms, and then unit-normalizing the data points before applying product quantization. When computing asymmetric distances, this simply requires a scalar multiplication of the PQ codebook once per scalar quantizer, which has negligible computational cost in practice.

To scale quantization based search techniques to massive datasets, a popular strategy is to first vector quantize the input vectors in the original space (coarse quantization), and then apply product quantization on the vector quantization residuals [20]. However, in such a 'VQ-PQ' style approach, the norms of the residuals exhibit significant variance. Therefore, the proposed multiscale approach provides significant gains for massive search even when the original data is fully normalized.

## 1.2 Related Works

The original idea of product quantization traces back to early works of signal processing [14, 12]. Jégou et al. [20] first introduced efficient asymmetric distance computation (ADC) and applied it to the approximate nearest neighbor (ANN) search problem. Since then, there have been multiple lines of work focused on improving PQ.

**Coarse Quantizer**. Also termed inverted file (IVF) indexing structure in Jégou et al. [20], this approach learns a vector quantization of the data points via clustering, using the cluster indices to form an inverted index storing all data points corresponding to a given cluster index consecutively. A data point is encoded via PQ codes associated with the residual (offset) of the data point from its closet cluster center. This design enables non-exhaustive search by searching only a subset of the $m$

clusters/partitions in the inverted index. However, previous works have learned coarse quantizers as a separate preprocessing step, without training the coarse quantizers jointly with the PQ codebooks.

**Rotation Matrix**. Since PQ quantizes each subspace independently, a rotation matrix can be applied to reduce the intra-subspace statistical dependence. Researchers have proposed multiple ways to estimate such a rotation matrix: Norouzi and Fleet [30] use ITQ [13] style alternating quantization; Optimized PQ [11] also applied a simple strategy to minimize the quantization error; Locally Optimized PQ [22] learns a separate $\mathbf{R}$ for each coarse partition (and incurs the extra overhead of multiplying each local rotation with the query vector to compute lookup tables specific to each partition). In high-dimensional setup, Zhang et al. [39] address the scalability issue in learning the $d \times d$ rotation matrix by imposing a Kronecker product structure. While learning such orthogonal transformations is a good strategy in general, it does not change the norm of data points. Thus it still suffers from norm variance as discussed in Section 1.1.

**Additive Codebooks**. Another line of research is focused on learning additive codebooks instead of subspace codebooks. This includes additive quantization [5, 6, 26], composite quantization [37, 38] and stacked quantization [27]. Since they do not work in subspaces, additive codebooks don't require rotation, although they are harder to learn and more expensive to encode. Empirically, such additive codebooks are more expressive, and outperform OPQ at lower bitrates. However, OPQ achieves similar performance at higher bitrates. Since additive codebooks don't address the variance of data point norms, the proposed multiscale approach can also be applied to additive codebooks as well.

**Implementation Improvements**. Much effort has been put into optimizing the implementation of ADC, as it is computationally critical. Douze et al. [10] propose using Hamming distance for fast pruning. Johnson et al. [21] come up with an efficient GPU implementation for ADC lookup. André et al. [2] propose to use SIMD-based computation to compute lower bounds for ADC. Our method is compatible with all of these improvements. We also discuss our ADC implementation in Section 4.4.

**Non-quantization Techniques**. There is a large body of similarity search literature on non-quantization based methods in both inner product search and nearest neighbor search. Tree based methods [7, 29, 9], graph based methods [16] and locality sensitive hashing style algorithms [19, 1, 32] focus on non-exhaustive search by partitioning the search space. In practice, these often lead to random memory accesses, and are often combined with exhaustive methods in ways similar to IVFADC [20, 4, 31, 28]. Binary embedding based approaches [36, 24, 18, 13, 17, 25] focus on learning short binary codes, and can be searched efficiently in Hamming space. However, there is typically a large gap between the precision of distance computations in Hamming vs. product codes under the same bitrate, and ADC can be computed with similar speed ([2, 21], Section 4.4). Therefore, we focus on comparison to ADC based techniques in this paper.

### 1.3 Contributions

We propose a complete *end-to-end* training algorithm to learn coarse quantizers, a rotation matrix, and product quantization codebooks, together with scalar quantizers to capture coarse quantization residual norms. This differs from prior work in that it (a) identifies and addresses the problem of variance in data point norms; (b) includes coarse quantizers as a part of the optimization; and (c) is *end-to-end* trainable using stochastic gradient descent (SGD), which leads to a significant improvement in quantization error compared to previous methods using alternating optimization [30, 11]. We also present ablation tests demonstrating the importance of each component of the algorithm in Section 4.2. In addition, we present theoretical motivation for our approach in Section 3.

## 2 Methodology

We focus on minimizing quantization error $\|x - \tilde{x}\|_2$, where $x$ is a data point and $\tilde{x}$ is its quantized approximation, as a proxy for minimizing query-database distance approximation error $|\|q - x\|_2 - \|q - \tilde{x}\|_2|$. State-of-the-art quantization techniques take a hierarchical approach [11, 27]. For instance, one or more "coarse" quantization stages (VQ) can be followed by product quantization (PQ) of the vector quantization residuals. A learned rotation is often applied to the residuals prior to product quantization to further reduce quantization error [11].

This style of approach provides two key benefits:

1. Real world data is often clusterable, with the diameter of clusters substantially lower than the diameter of the dataset as a whole. The vector quantization can thus be used to obtain a "residual dataset" with much smaller diameter, yielding significant reductions in quantization error when quantized with only a product code [15].

2. By additionally learning a rotation of the VQ residuals, the variance within each PQ subspace can be significantly reduced for many real world datasets, yielding substantially lower quantization error and correspondingly higher recall.

As noted in Section 1.1, an additional source of quantization error when performing product quantization is the variance of data point norms. We extend the above strategy by explicitly representing the norm of VQ residuals, learning a PQ codebook only on the *unit-normalized* rotated VQ residuals, while separately scalar quantizing the residual norm scales. Specifically, multiscale quantization employs the following steps: (1) vector quantization of the dataset; (2) learned rotation of the vector quantization residuals; (3) reparameterization of the rotated residuals into direction and scale components; (4) product quantization of the direction component; (5) scalar quantization of the scale component.

Formally, in multiscale quantization, the rotated residual $r_x$ and its $\ell_2$ normalized version $\hat{r}_x$ are defined as:

$$r_x = \mathbf{R}(x - \phi_{VQ}(x)), \quad \hat{r}_x = r_x/\|r_x\|_2$$

And a data point $x \in \mathbb{R}^d$ is approximated by

$$x \approx \tilde{x} = \phi_{VQ}(x) + \tilde{r}_x, \text{ where } \tilde{r}_x = \phi_{SQ}(\lambda_x)\mathbf{R}^T\phi_{PQ}(\hat{r}_x) \text{ and } \lambda_x = \|r_x\|_2/\|\phi_{PQ}(\hat{r}_x)\|_2 \quad (2)$$

From above, $\phi_{VQ}(x) = \mathrm{argmin}_{c \in \{\mathbf{C}_j\}} \|x - c\|_2$ returns the closest vector quantization codeword for $x$; $\mathbf{C} \in \mathbb{R}^{d \times m}$ is a vector quantization codebook with $m$ codewords; $\mathbf{C}_j$ is its $j$-th codeword (i.e. the $j$-th column of $\mathbf{C}$); And the matrix $\mathbf{R} \in \mathbb{R}^{d \times d}$ is a learned rotation, applied to the residuals of vector quantization; The residual norm scale $\lambda_x$ is a scalar multiplier to the product quantized $\phi_{PQ}(\hat{r}_x)$ that helps preserve the $\ell_2$ norm of the rotated residual $r_x$; And $\phi_{SQ}$ returns the nearest scalar quantizer from a scalar quantization codebook $\mathbf{W} \in \mathbb{R}^p$ with $p$ codewords (equivalent to one-dimensional vector quantization). The product quantizer $\phi_{PQ}(r_x)$ is given by

$$\phi_{PQ}(\hat{r}_x) = \begin{pmatrix} \phi_{PQ}^{(1)}(\hat{r}_x^{(1)}) \\ \phi_{PQ}^{(2)}(\hat{r}_x^{(2)}) \\ \vdots \\ \phi_{PQ}^{(K)}(\hat{r}_x^{(K)}) \end{pmatrix}, \quad \hat{r}_x = \begin{pmatrix} \hat{r}_x^{(1)} \\ \hat{r}_x^{(2)} \\ \vdots \\ \hat{r}_x^{(K)} \end{pmatrix}$$

as the concatenation of codewords obtained by dividing the rotated and normalized residuals $\hat{r}_x$ into $K$ subvectors $\hat{r}_x^{(k)}$, $k = 1, 2, \cdots, K$, and quantizing the subvectors independently by vector quantizers $\phi_{PQ}^{(k)}(\cdot)$ to minimize quantization error:

$$\phi_{PQ}^{(k)}(\hat{r}_x^{(k)}) = \mathrm{argmin}_{s \in \{\mathbf{S}_j^{(k)}\}} \|\hat{r}_x^{(k)} - s\|_2.$$

Hence, $\mathbf{S}^{(k)} \in \mathbb{R}^{d^{(k)} \times l}$ is the vector quantization codebook for the $k$-th subspace (with $l$ codewords). Frequently, $d^{(k)}$, the dimension of the $k$-th subvector, is simply $\frac{d}{K}$, although subvectors of varying size are also possible.

The quantized, normalized residuals are represented by the $K$ indices of index($\phi_{PQ}^{(k)}(\hat{r}_x^{(k)})$), $k = 1, \cdots, K$. This representation has an overall bitrate of $K \log_2 l$, where $K$ is the number of subspaces, and $l$ is the number of product quantizers in each subspace. The residual norm scales are maintained by organizing the residuals associated with a VQ partition into groups, where within a group all residuals have the same quantized norm scale. The groups are ordered by quantized norm scale, and thus only the indices of group boundaries need to be maintained. The total storage cost including group boundaries and scalar quantization levels is thus $O(mp)$, where $m$ is number of vector quantizers and $p$ is the number of scalar quantizers. In our experiments, we set $p$ to 8, which we find has a negligible effect on recall compared with using unquantized norm scales.

## 2.1 Efficient Search under Multiscale Quantization

The multiscale quantization model enables nearest neighbor search to be carried out efficiently. For a query $q$, we compute the squared $\ell_2$ distance of $q$ with each codeword in the vector quantization codebook $\mathbf{C}$, and search further within the nearest VQ partition. Suppose the corresponding quantizer is $c_q^* = \operatorname{argmin}_{c \in \{\mathbf{C}_j\}} \|q - c\|_2$, and the corresponding quantization partition is $P_q^* = \{x \in \{X_j\}_{[N]} \,|\, \phi_{VQ}(x) = c_q^*\}$. Then, the approximate squared $\ell_2$ distance between the query and database points in $P_q^*$ are computed using a lookup table. The final prediction is made by taking the database point with the smallest approximate distance, i.e.

$$x_q^{\text{pred}} = \operatorname*{argmin}_{x \in P_q^*} \left( \|q - c_q^*\|_2^2 - 2 \left[ \mathbf{R}(q - c_q^*) \right] \cdot [\phi_{SQ}(\lambda_x)\phi_{PQ}(\hat{r}_x)] + \|\phi_{SQ}(\lambda_x)\phi_{PQ}(\hat{r}_x)\|_2^2 \right).$$

We use a lookup table to compute the quantized inner product between subvectors of the query's rotated VQ residual $\bar{q} = \mathbf{R}(q - c_q^*)$ and the scaled product quantized data point residuals $\phi_{SQ}(\lambda_x)\phi_{PQ}(\hat{r}_x)$. Letting $\bar{q}^{(k)}$ be the $k$-th subvector of $\bar{q}$ and $w_x = \phi_{SQ}(\lambda_x)$ the quantized norm scale, we first precompute inner products and the squared quantized $\ell_2$ norm with the PQ codebook $\mathbf{S}$ as $v_j^{(k)} = -2\bar{q}^{(k)} \cdot w_x \mathbf{S}_j^{(k)} + w_x^2 \|\mathbf{S}_j^{(k)}\|_2^2$ for all $j$ and $k$, giving $K$ lookup tables $v^{(1)}, \ldots, v^{(K)}$ each of dimension $l$. We can then compute

$$-2\bar{q} \cdot w_x \phi_{PQ}(r_x) + w_x^2 \|\phi_{PQ}(r_x)\|_2^2 = \sum_{k=1}^{K} v_{\text{index}(\phi_{PQ}^{(k)}(r_x))}^{(k)}$$

In practice, instead of searching only one vector quantization partition, one can use soft vector quantization and search the $t$ partitions with the lowest $\|q - \mathbf{C}_j\|_2$. The final complexity of the search is $\mathcal{O}(\frac{NtK}{m})$.

In our implementation, since all the data points with the same quantized norm scale are stored in consecutive groups, we need only create a new lookup table at the beginning of a new group, by combining scale independent lookup tables of $-2\bar{q}^{(k)} \cdot \mathbf{S}_j^{(k)}$ and $\|\mathbf{S}_j^{(k)}\|_2^2$ (multiplied by $w_x$ and $w_x^2$, respectively) using hardware optimized fused multiply-add instructions. We incur this computation cost only $p$ times for a VQ partition, where $p$ is the number of scalar quantizers. In our experiment, we set $p = 8$ and the number of VQ partitions to search $t = 8$, maintaining relatively low performance overhead. We discuss more on the lookup table implementation in Section 4.4.

## 2.2 Optimization Procedure

We can explicitly formulate the mean squared loss as a function of our parameter vector $\mathbf{\Theta} = (\mathbf{C}; \{\mathbf{S}^{(k)}\}_{[K]}; \mathbf{R}; \{\mathbf{W}_i\}_{[m]})$ per our approximation formulation (2). $\mathbf{W}_i$ here represents the parameter vector for the scalar quantizer of norm scales in partition $i$. To jointly train the parameters of the model, we use stochastic gradient descent. To optimize the orthogonal transformation of vector quantization residuals while maintaining orthogonality, we parameterize it via the Cayley characterization of orthogonal matrices [8]:

$$\mathbf{R} = (\mathbf{I} - \mathbf{A})(\mathbf{I} + \mathbf{A})^{-1}, \tag{3}$$

where $\mathbf{A}$ is a skew-symmetric matrix, i.e. $\mathbf{A} = -\mathbf{A}^T$. Note that (3) is differentiable w.r.t. the $d(d-1)/2$ parameters of $\mathbf{A}$. Computing the gradient requires an inversion of a $d \times d$ matrix at each iteration. However we found this tradeoff to be acceptable for datasets with dimensionalities in the hundreds to thousands. When applying this method on high-dimensional datasets, one can restrict the number of parameters of $\mathbf{A}$ to trade off capacity and computational cost.

The codebook for vector quantization is initialized using random samples from the dataset, while the codebook for product quantization is initialized using the residuals (after vector quantization, normalization and rotation) of a set of independent samples. To allow the vector quantization a chance to partition the space, we optimize only the vector quantization error for several epochs before initializing the product codes and doing full joint training. The parameters of the skew-symmetric matrix $\mathbf{A}$ were initialized by sampling from $\mathcal{N}(0, 1e-3)$.

All optimization parameters were fixed for all datasets (although we note it would be possible to improve results slightly with more extensive per-dataset tuning). We used the Adam optimization algorithm [23] with the parameters suggested by the authors, minibatch sizes of 2000, and a learning rate of $1e-4$ during joint training (and $1e-3$ when training only the vector quantizers).

To learn the scalar quantizer for residual norm scales and capture their local distribution within a VQ partition, we jointly optimize the assignment of PQ codes and the scalar quantizer for all data points within the same partition. Leaving the PQ codebook and rotation fixed, we alternate between following two steps until convergence:

1. Fix all assigned PQ codes and scalar quantize the norm scales $\lambda_x = \|r_x\|_2/\|\phi_{PQ}(\hat{r}_x)\|_2$ *only* within the partition.

2. Fix all quantized norm scales within the partition and reassign PQ codes for $r_x/\phi_{SQ}(\lambda_x)$.

In practice, it only takes a few iterations to converge to a local minimum for every VQ partition.

## 3 Analysis

Below we provide theoretical motivation and analysis for the components of the proposed quantization approach, including for multiscale, learned rotation, and coarse quantization stages.

### 3.1 Multiscale

We first show that adding a scalar quantizer further increases the recall when the norms of the residuals exhibit large variance. For a query $q$ and a given partition with center $\mathbf{C}_j$, if we define $q_j = q - \mathbf{C}_j$, then the $\ell_2$ error caused by residual quantization is

$$\left|\|q_j - r_x\|_2^2 - \|q_j - \tilde{r}_x\|_2^2\right| = \left|-2q_j \cdot (r_x - \tilde{r}_x) + \|r_x\|_2^2 - \|\tilde{r}_x\|_2^2\right|$$
$$\leq |2q_j \cdot (r_x - \tilde{r}_x)| + \left|\|r_x\|_2^2 - \|\tilde{r}_x\|_2^2\right|.$$

The first query dependent term can be further transformed as

$$|2q_j \cdot (r_x - \tilde{r}_x)| = 2\sqrt{[(r_x - \tilde{r}_x)^T q_j][q_j^T(r_x - \tilde{r}_x)]} = 2\sqrt{(r_x - \tilde{r}_x)^T(q_j q_j^T)(r_x - \tilde{r}_x)}$$

Taking expectation w.r.t $q$ yields

$$\mathbb{E}_q|2q_j \cdot (r_x - \tilde{r}_x)| \leq 2\sqrt{\mathbb{E}_q[(r_x - \tilde{r}_x)^T(q_j q_j^T)(r_x - \tilde{r}_x)]} = 2\sqrt{(r_x - \tilde{r}_x)^T\mathbb{E}_q(q_j q_j^T)(r_x - \tilde{r}_x)},$$

where the inequality follows from Jensen's inequality. If $\lambda_q$ is the largest eigen value of the covariance matrix $\mathbb{E}_q(q_j q_j^T)$, then

$$\mathbb{E}_q\left|\|q_j - r_x\|_2^2 - \|q_j - \tilde{r}_x\|_2^2\right| \leq 2\sqrt{\lambda_q}\|r_x - \tilde{r}_x\|_2 + \left|\|r_x\|_2^2 - \|\tilde{r}_x\|_2^2\right|.$$

Existing quantization methods have focused on the first term in the error of $\ell_2$ distance. However for VQ residuals with large variance in $\|r_x\|_2$, the second quadratic term becomes dominant. By scalar quantizing the residual norm scales, especially within each VQ partition locally, we can reduce the second term substantially and thus improve recall on real datasets.

### 3.2 Rotation Matrix

Performing quantization after a learned rotation has been found to work well in practice [13, 30]. Here we show rotation is required in some scenarios. Let $x_i = \mathbf{R}y_i, 1 \leq i \leq n$. We show that there exist simple examples, where the $y_i$'s have a product code with small codebook size and MSE 0, whereas to get any small MSE on $x_i$s one may need to use exponentially many codewords. On real-world datasets, this difference might not be quite so pronounced, but it is still significant and hence undoing the rotation can yield significantly better MSE. We provide the following Lemma (see the supplementary material for a proof).

**Lemma 1.** *Let* $\mathbf{X} = \mathbf{RY}$, *i.e., for* $1 \leq i \leq n$, $x_i = \mathbf{R}y_i$. *There exists a dataset* $\mathbf{Y}$ *and a rotation matrix* $\mathbf{R}$ *such that a canonical basis product code of size* 2 *is sufficient to achieve MSE of* 0 *for* $\mathbf{Y}$, *whereas any product code on* $\mathbf{X}$ *requires* $2^{c \cdot \min(d/K, K)\epsilon}$ *codewords to achieve MSE* $\epsilon\|x\|_{\max}$, *where* $c$ *is some universal constant and* $\|x\|_{\max}$ *is the maximum* $\ell_2$ *norm of any data point.*

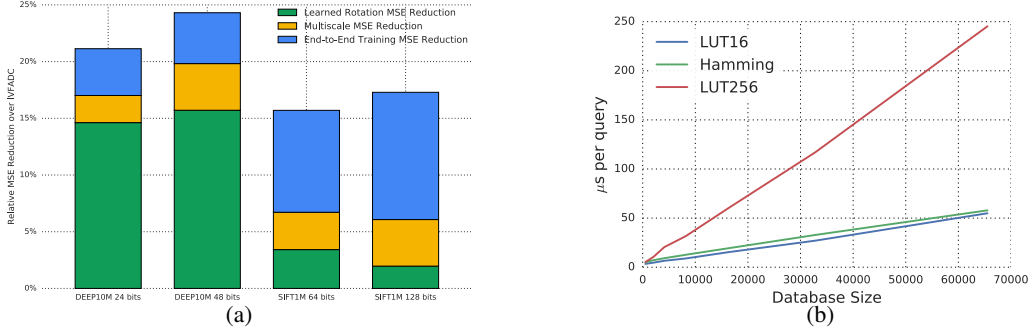

(a)  (b)

Figure 2: (a) Break down by contribution to MSE reduction from each component in our model on SIFT1M and DEEP10M datasets with different bitrates. The baseline is the original IVFADC setup with no rotation or norm scale quantization. (b) Time spent per query by different distance computation methods on linear search of a database of size $|\mathbf{X}| = 2^7, 2^8, 2^9, \cdots 2^{16}$ under 128 bits. Lower curves indicate faster search time.

## 3.3 Coarse Quantization

We analyze the proposed vector and product quantization when the data is generated by a $K$-subspace mixture model that captures two properties observed in many real-world data sets: samples belong to one of several underlying categories, also referred to as components, and within each component the residuals are generated independently in $K$ subspaces. The precise model is defined in Appendix B.

For a query $q$, let $x_q^*$ be the sample that minimizes $\|q - x\|_2$. Let $x_q^{VQ}$ be the output of the hierarchical nearest neighbor algorithm that first finds the nearest cluster center and then searches within that cluster. We show that if $q$ is generated independently of $x$, then with high probability it returns an $x_q^{VQ}$ that is near-optimal.

**Theorem 1.** *Given $n$ samples from an underlying $K$-subspace mixture model that has been clustered correctly and an independently generated query $q$, with probability $\geq 1 - \delta$,*

$$\left| \|q - x_q^*\|_2^2 - \|q - x_q^{VQ}\|_2^2 \right| \leq 8b\sqrt{\frac{dr^2}{2K}\log\frac{4n}{\delta}} + 4r^2\sqrt{\frac{d^2}{2K}\log\frac{2n}{\delta}}.$$

See Appendix B for a proof. Note that $r = \max_{x \in X}\|r_x\|_\infty$ is the maximum value of the residual in any coordinate and offers a natural scaling for our problem and $b = \max_{x \in X}\|q - x\|_2$ is the maximum distance between $q$ and any data point.

## 4 Experiments

### 4.1 Evaluation Datasets

We evaluate the performance of end-to-end trained multiscale quantization (MSQ) on the SIFT1M [20] and DEEP10M [3] datasets, which are often used in benchmarking the performance of nearest neighbor search. SIFT1M [20] contains 1 million, 128 dimensional SIFT descriptors extracted from Flickr images. DEEP10M is introduced in [3], by extracting 96 PCA components from the final hidden layer activations of GoogLeNet [33].

At training time, each dataset is indexed with 1024 VQ coarse quantizers. At query time, quantized residuals from the 8 partitions closest to the query are further searched using ADC to generate the final nearest neighbors. We report results on both quantization error (MSE, Section 4.2) and in terms of retrieval recall (Recall1@N, Section 4.3). Often, the two metrics are strongly correlated.

### 4.2 Ablation Tests

Compared to IVFADC [20], which uses plain PQ with coarse quantizers, our end-to-end trained MSQ reduces quantization error by 15-20% on SIFT1M, and 20-25% on DEEP10M, which is a substantial reduction. Multiple components contribute to this reduction: (1) learned rotation of the VQ residuals; (2) separate quantization of the residual norms into multiple scales; and (3) end-to-end training of all parameters.

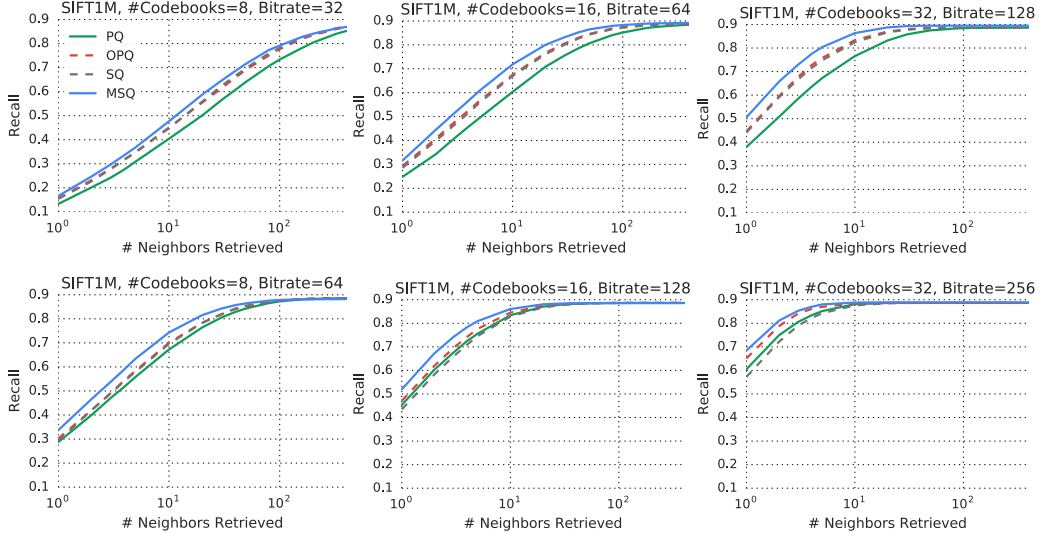

Figure 3: Recall curves when retrieving Top-1 neighbors (`Recall1@N`) on the SIFT1M dataset with varying numbers of codebooks and centers. We search $t = 8$ out of $m = 1024$ VQ partitions.

In order to understand the effect of each component, we plot the MSE reduction relative to IV-FADC [20] for several ablation tests (Figure 2a). On DEEP10M, the proposed multiscale approach and the end-to-end learning contribute an additional 5-10% MSE reduction on top of learned rotation, while they contribute 10-15% on SIFT1M. It is important to note that on SIFT1M, multiscale quantization and end-to-end training have a bigger impact than learned rotation, which is itself often considered to yield a significant improvement.

## 4.3 Recall Experiments

We compare the proposed end-to-end trained multiscale quantization method against three baselines methods: *product quantization (PQ)* [20], *optimized product quantization (OPQ)* [11] and *stacked quantizers (SQ)* [27]. We generate ground-truth results using brute force search, and compare the results of each method against ground-truth in fixed-bitrate settings.

For fixed-bitrate experiments, we show recall curves for varying numbers of PQ codebooks from the range $\{8, 16, 32\}$ for the SIFT1M dataset and $\{6, 12, 24\}$ for the DEEP10M dataset. For each number of codebooks, we experimented with both 16 centers for in-register table lookup and 256 centers for in-memory table lookup in Figure 3 and 4. From the recall curves, it is clear that multiscale quantization performs better than all baselines across both datasets in all settings.

## 4.4 Speed Benchmarks

We use the same indexing structure (IVF), and the same ADC computation implementation for all baselines (PQ [20], OPQ [11], SQ [27]). Thus the speed of all baselines are essentially identical at the same bitrate, meaning Figure 3 and 4 are both fixed-memory and fixed-time, and thus directly comparable. For codebooks with 256 centers, we implemented in-memory lookup table (LUT256) [20]; for codebooks with 16 centers, we implemented in-register lookup table (LUT16) using the `VPSHUFB` instruction from `AVX2`, which performs 32 lookups in parallel.

Also, we notice that there have been different implementations of ADC. The original algorithm proposed in [20] uses in-memory lookup tables. We place tables in SIMD registers and leverage SIMD instructions for fast lookup. Similar ideas are also reported in recent literature [10, 17, 2]. Here we put them on equal footing and provide a comparison of different approaches. In Figure 2b, we plot the time for distance computation at the same bitrate. Clearly, `VPSHUFB` based LUT16 achieves almost the same speed compared to `POPCNT` based Hamming, and they are both 5x faster than in-memory based ADC. As a practical observation, when the number of neighbors to be retrieved is large, `Recall1@N` of LUT256 and LUT16 is often comparable at the same bitrate, and LUT16 with 5x speed up is almost always preferred.

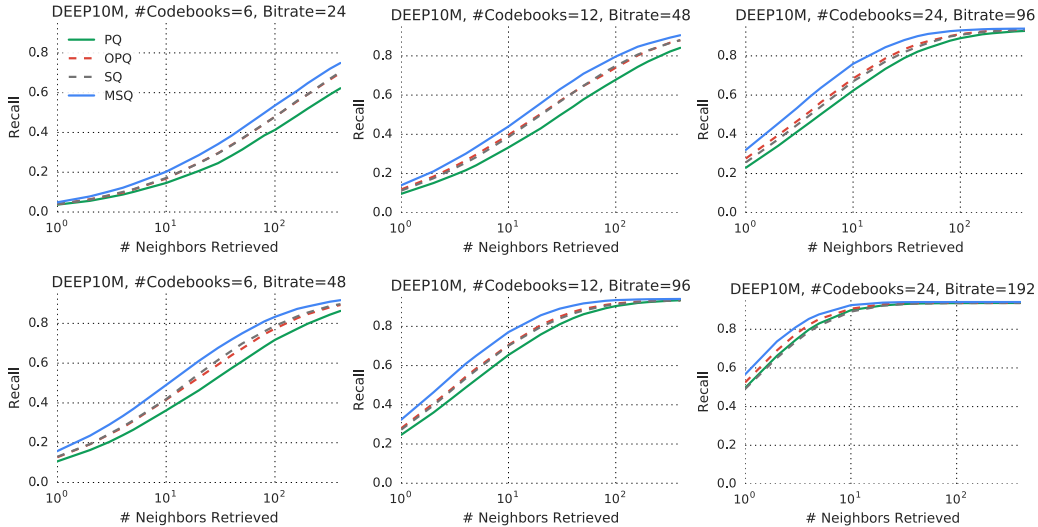

Figure 4: Recall curves when retrieving Top-1 neighbors (`Recall1@N`) on the DEEP10M datasets varying numbers of codebooks and centers. We search $t = 8$ out of $m = 1024$ VQ partitions.

## 5 Conclusions

We have proposed an *end-to-end* trainable multiscale quantization method that minimizes overall quantization loss. We introduce a novel scalar quantization approach to account for the variances in data point norms, which is both empirically and theoretically motivated. Together with the end-to-end training, this contributes to large reduction in quantization error over existing competing methods that already employ optimized rotation and coarse quantization. Finally, we conducted comprehensive nearest neighbor search retrieval experiments on two large-scale, publicly available benchmark datasets, and achieve considerable improvement over state-of-the-art.

## 6 Acknowledgements

We thank Jeffrey Pennington and Chen Wang for their helpful comments and discussions.

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
