[Supplementary Material]

## Appendix A Proof of Lemma 1

*Proof.* Let $s = \min(d/K, K)$ and $n = 2^s$. Let $v$ be an $s$ dimensional vector. Let $y(v)$ be a vector such that for all $1 \le i \le s, 1 \le j \le s, y((i-1)d/K + j) = v(i)$ and the remaining entries are 0. Let **Y** be the set of all such vectors corresponding to $v \in \{-1, 1\}^s$:

$$\mathbf{Y} = \{y(v) : v \in \{-1, 1\}^s\}.$$

We first note that for all points in **Y** and for all $1 \le i \le K$, $[y((i-1)d/K + 1), y((i-1)d/K + 2), y((i-1)d/K + 1), \ldots, y((i-1)d/K + d/K)]$ takes one of two values and hence **Y** has a canonical basis product code with codebook size 2.

Let **R** be a permutation matrix such that for all $1 \le i \le s$ and $1 \le j \le s$

$$y(id/K + j) \to y((i + j - 1)d/K + j \bmod (d)).$$

This ensures that within each subspace, $1 \le i \le s$, $[x((i-1)d/K + 1), x((i-1)d/K + 2), x((i-1)d/K + 1), \ldots, x((i-1)d/K + s)]$ takes all possible $2^s$ possibilities. Hence, by the Gilbert-Varshamov bound, any product code to achieve a mean squared loss of $\epsilon\|x\|_2$ requires at least $2^{c \cdot s\epsilon}$ codewords to achieve a worst case MSE of $\epsilon\|x\|_2$. $\square$

## Appendix B Coarse quantization

To define the exact K-subspace mixture model, we need few definitions.

Let $D^{(1)}, D^{(2)}, \ldots D^{(K)}$ be $K$ be subspaces of equal dimension that span $\mathbf{R}^d$. We call a probability distribution $p$ *K-subspace independent* if the projections of the sample on $D^{(1)}, D^{(2)}, \ldots D^{(K)}$ are statistically generated according to $K$ independent probability distributions $p^{(1)}, p^{(2)}, \ldots p^{(K)}$.

We define a *K-subspace mixture model* as follows. Each sample $X$ is generated as

$$X = \mathbf{R}(\mu + Y),$$

where $R$ is an arbitrary but fixed rotation matrix, $\mu$ is the mean of components which can take one of $m$ values $\mu_1, \mu_2, \ldots \mu_m$, and $Y$ is the *residual* which is generated according to a $K$-subspace independent distribution. Without loss of generality we assume that $Y$ is a zero mean random variable.

As stated before, the above model captures several phenomena such as *clusterability* and *low-dimensionality* of the underlying data within each subspace. Furthermore, it allows us to use product codes even when data is arbitrarily rotated. Finally, if we cluster the data, then the MSE of the product codes depends only on the radius of the residuals as opposed to the whole data. Thus, stronger guarantees can be found using previous results on the MSE of product codes [15].

*Proof of Theorem 1.* The proof relies on the following lemmas, which is a direct application of McDiarmid's inequality.

**Lemma 2.** *Let $Y$ be generated according to a zero-mean $K$-subspace independent distribution such that within each coordinate, the maximum value of $Y$ is at most $r$. Then for a $q$,*

$$Pr\left(|q^T Y| \ge t\right) \le 2e^{\frac{-2t^2}{\|q\|_2^2 dr^2/K}}.$$

*Proof.* Recall that for a vector $x$, $x^{(k)}$ denotes its component along subspace $D^{(k)}$. Observe that

$$q^T Y = \sum_{k=1}^{K} (q^{(k)})^T Y^{(k)}.$$

Since $Y$ is generated according to a $K$-subspace independent distribution, each of the $Y^{(k)}$ are independent. Changing $Y^{(k)}$ changes $q^T Y$ by at most $\|q^{(k)}\|_2 \cdot \sqrt{dr^2/K}$. Thus, by McDiarmid's inequality,

$$\Pr\left(|q^T Y| \ge t\right) \le 2e^{\frac{-2t^2}{\|q\|_2^2 dr^2/K}}.$$

$\square$

**Lemma 3.** *Let $Y$ be generated according to a zero-mean $K$-subspace independent distribution such that within each coordinate, then*

$$Pr\left(\|Y\|_2^2 \geq \mathbb{E}[\|Y\|_2^2] + t\right) \leq e^{\frac{-2Kt^2}{d^2r^4}}.$$

*Proof.* Recall that for a vector $x$, $x^{(k)}$ denotes its component along subspace $D^{(k)}$. Since $Y$ is generated according to a $K$-subspace independent distribution, each of the $Y^{(k)}$ are independent. Changing $Y^{(k)}$ changes $\|Y\|_2^2$ by at most $dr^2/K$. Thus, by McDiarmid's inequality,

$$\text{Pr}\left(\|Y\|_2^2 \geq \mathbb{E}[\|Y\|_2^2] + t\right) \leq e^{\frac{-2Kt^2}{d^2r^4}}.$$

$\square$

Let $\mu_1$ be the cluster center corresponding to $x_q^*$ and $\mu_2$ be the cluster center corresponding to $x_q^{VQ}$. Let $\tilde{\mu}_1$ and $\tilde{\mu}_2$ be their estimates. Let $y_q^{VQ} = x_q^{VQ} - \mu_2$ and $y_q^* = x_q^* - \mu_1$. If $\mu_1 = \mu_2$, then our algorithm outputs $x_q^*$ and hence $x_q^{VQ} = x_q^*$. If $\mu_1 \neq \mu_2$, note that

$$\|q - x_q^*\|_2^2 - \|q - x_q^{VQ}\|_2^2$$
$$= \|q - (\mu_1 + y_q^*)\|_2^2 - \|q - (\mu_2 + y_q^{VQ})\|_2^2$$
$$= \|q - \mu_1\|_2^2 - \|q - \mu_2\|_2^2 + \|y_q^*\|_2^2 - \|y_q^{VQ}\|_2^2 + 2(q - \mu_1)^T y_q^* - 2(q - \mu_2)^T y_q^{VQ}$$
$$= \|q - \tilde{\mu}_1\|_2^2 - \|q - \tilde{\mu}_2\|_2^2$$
$$\quad + 2(q - \mu_1)^T y_q^* - 2(q - \mu_2)^T y_q^{VQ} + 2(q - \tilde{\mu}_1)^T (\tilde{\mu}_1 - \mu_1) - 2(q - \tilde{\mu}_2)^T (\tilde{\mu}_2 - \mu_2)$$
$$\quad + \|\mu_1 - \tilde{\mu}_1\|_2^2 - \|\mu_2 - \tilde{\mu}_2\|_2^2 + \|y_q^*\|_2^2 - \|y_q^{VQ}\|_2^2$$
$$\overset{(a)}{\leq} 2(q - \mu_1)^T y_q^* - 2(q - \mu_2)^T y_q^{VQ} + 2(q - \tilde{\mu}_1)^T (\tilde{\mu}_1 - \mu_1) - 2(q - \tilde{\mu}_2)^T (\tilde{\mu}_2 - \mu_2)$$
$$\quad + \|\mu_1 - \tilde{\mu}_1\|_2^2 - \|\mu_2 - \tilde{\mu}_2\|_2^2 + \|y_q^*\|_2^2 - \|y_q^{VQ}\|_2^2$$
$$\overset{(b)}{\leq} 8 \max_{i:\|q\|_2^2 \leq b} q^T y_i + \|\mu_1 - \tilde{\mu}_1\|_2^2 - \|\mu_2 - \tilde{\mu}_2\|_2^2 + \|y_q^*\|_2^2 - \|y_q^{VQ}\|_2^2$$
$$\overset{(c)}{\leq} 8 \max_{i:\|q\|_2^2 \leq b} q^T y_i + 4 \max_i |\|y_i\|_2^2 - \mathbb{E}[\|Y_i\|_2^2]|.$$

where $(a)$ follows from the fact that the algorithm favors $\tilde{\mu}_2$ over $\tilde{\mu}_1$ and hence $q^T \tilde{\mu}_2 \geq q^T \tilde{\mu}_1$. $(b)$ and $(c)$ follow from triangle inequality and the fact that $\mu_1 - \tilde{\mu}_1$ is the average of $y_i$s in that cluster. By Lemmas and 3, with probability $\geq 1 - \delta$, for all $n$ samples,

$$8 \max_{i:\|q\|_2^2 \leq b} q^T y_i + 4 \max_i |\|y_i\|_2^2 - \mathbb{E}[\|Y_i\|_2^2]| \leq 8b\sqrt{\frac{dr^2}{2K} \log \frac{4n}{\delta}} + 4r^2 \sqrt{\frac{d^2}{2K} \log \frac{2n}{\delta}}.$$

and hence the result.

$\square$