[Reviews · NeurIPS 2017]

Reviewer 1



In this paper, the authors propose a new quantization method to improve the ANN search performance. As an extension of Product Quantization (PQ), the method fixes the drawbacks of PQ when dealing with data with high norm variance. The motivation is clear and meaningful. And experiments on two large datasets are conducted to show the effectiveness of the proposed method. To solve the drawbacks of PQ, the authors make several modifications of PQ: (1) learn a rotation to the VQ residuals which reduces the variance (2) add a scalar quantizier to explicitly represent the norm of VQ residuals (3) optimize the objective function in an end-to-end way As is shown in experiments, all these aspects contribute to the final improvement. However, these three modifications seem to contribute differently on different datasets. What is the reason behind this? (I notice that Deep Learning features are reduced by PCA, why doing this?). I am also curious about the result on more datasets such as GIST-1M. And there are also some aspects that are not covered by this paper. (1) What about doing some rotation (full PCA or learned) to the original data? PQ seems to achieve better results. (2) What if a scalar quantizier is also added to represent VQ? Will this also improve the final result?

Reviewer 2



This paper proposes an end-to-end trainable multiscale quantization method that minimizes overall quantization loss. A novel scalar quantization approach is proposed to account for the variances in datapoint norms. The motivation seems clear. Based on the end-to-end training, this largely reduces the quantization error over existing competing methods that already employ optimized rotation and coarse quantization. This method is evaluated (based on the performance of end-to-end trained multi scale quantization) on the SIFT-1M [19] and DEEP-10M [3] datasets. It is mainly compared with baselines (PQ [19], OPQ [11], SQ [26]), and shows better performance. Overall this paper is well written. The motivation is clear and the solution seems reasonable, with several evaluations to support this motivation. One main concern is that this method is mainly compared with [19] (PAMI 2011), [11] (PAMI 2014), [26] (CoRR abs 2014). However, this is usually not sufficient for the experimental evaluation. State of the art (and more recent) hashing and/or quantization methods could be compared. Otherwise it is not easy to judge the effectiveness of this proposed method (e.g., accuracy or precision) among previous methods. So it would be better to have several more methods and recent methods, even they may use different types of algorithm framework. Also a minor question: "In practice, this takes only a few iterations to converge for every VQ partition." Possible to analyze more for the alternating optimization process? Possible that this converges to local minimum quickly? Is it sensitive to initializations? This might be an important factor for reproducing the results.

Reviewer 3



This paper proposes a framework called multiscale (vector) quantization (MSQ) for fast similarity search in massive datasets, motivated by the observation that the query-based quantization performance in a database is in fact affected by the norm of the residual vector after quantization. The authors then proposed MSQ to better quantize a vector based on the residual norms under a K-dimensional subspace model, given in the formulation of equation (2). The authors also provided some theoretical justifications (however, they are loose-ended and may not properly justify the proposed MSQ method), and showed improved performance when compared with three existing methods on SIFT-10M and DEEP-10M datasets. Although the experimental results seem promising by outperforming the compared methods, there are several issues that downgrade the contribution of this work, especially in the analysis part. The details are as follows. 1. Loose-ended analysis - The analysis in Section 3 does not properly justify the proposed MSQ method and some statements are problematic. 1) Lemma 1: The statement that "there exists a dataset Y and a rotation matrix R..." is a strange one. Shouldn't the statement be "For any dataset Y (such that some constraints apply)...". The current statement does not guarantee applicability to any considered datasets. 2) Theorem 1: By the definition, $x_q^*$ is a minimizer of $\|q-x\|_2$ for a given q. This implies the first term in the LHS is no greater than the second term, so a trivial upper bound on the LHS should be 0. In this context, I believe the authors are aiming for an upper bound on the difference of "second term - first term", instead of the current one. 3) Equation under Sec. 2.1 - seem to be able to be simplified by taking out the first term (independent of x). 2. Presentation and notation - This work is deeply motivated by the observation that the variance in the data points plays a key role in quantization performance, as demonstrated by Fig.1. However, the precise definition of a residual vector has never been explicitly defined (not to mention the existence of a rotation matrix). One can pretty much "guess" from equation (2) that r_x denotes the rotated residual vector, but the lack of precise definition can be confusing. In addition, the authors use the notation of general norm ($\|x\|$) and L2 norm ($\|x\|_2$) alternatively. While the authors are making efforts to make MSQ as general as possible (by allowing general norm), many important statements of the paper, including query distance approximation error and Theorem 1, are tied up with L2 norm. The experiments seemed to be assuming L2 norm setting as well. How generalizable is the proposed MSQ to general norm has not been made clear.

Reviewer 4



The paper proposes a novel approach to reduce quantization error in the scenario where the product quantization is applied to the residual values that are the result of the coarse whole-space quantization. To encode residuals the authors propose to (1) quantize the norm separately (2) apply product quantization to unit-normalized residuals. It trains everything in the end-to-end manner, which is certainly good. Pros: 1. I very much like the approach. 2. The paper is well-written. 3. A good coverage of work related to quantization (including several relevant recent papers). However, not all work related to efficient k-NN search (see below)! Cons: 1. The related work on non-quantization techniques doesn't mention any graph-based techniques. Compared to quantization approaches they are less SPACE-efficient (but so are LSH methods, even more so!), but they are super-fast. Empirically they often greatly outperform everything else, in particular, they outperform LSH. One recent relevant work (but there are many others): Harwood, Ben, and Tom Drummond. "FANNG: fast approximate nearest neighbour graphs." Proceedings of the IEEE Conference on Computer Vision and Pattern Recognition. 2016. I would encourage authors to mention this line of work, space permitting. 2. The paper is a bit cryptic. See, e.g., my comment to the recall experiments section. A few more detailed comments: I think I can guess where formula 4:158 comes from, but it won't harm if you show a derivation. 7: 280 What do you mean by saying that VPSHUFB performs lookups? It's a shuffle instruction, not a store/gather instruction. Recall experiments: Does the phrase "We generate ground-truth results using brute force search, and compare the results of each method against ground-truth in fixed-bitrate settings." mean that you do not compute accuracy for the k-NN search using *ORIGINAL* vectors? Please, also clarify how do you understand ground-truth in the fixed-bitrate setting. Figure 4: please, explain better the figures. I don't quite get what is exactly the number of neighbors on x axis. Is this the number of neighbors retrieved using an approximated distance, i.e., using the formula in 4:158? Speed benchmarks are a bit confusing, because these are only for sequential search using a tiny subset of data likely fitting into a L3 cache. I would love to see actually full-size tests. Regarding the POPCNT approach: what exactly do you do? Is it a scalar code? Is it single-thread? I am asking b/c I am getting twice as longer processing times using POPCNT for the Hamming distance with the same number of bits (128). What is your hardware: CPU, #of cores, RAM, cache size? Are your speed-benchmarks single-threaded? Which compiler do you use? Please, clarify if your approach to learning an orthogonal transformation (using Cayley factorization) is novel or not.